# Impact of Urbanization on the Environmental Regulation Efficiency in the Yangtze River Basin Based on the Empirical Analysis of Spatial Econometrics

**DOI:** 10.3390/ijerph18179105

**Published:** 2021-08-28

**Authors:** Qian Zhang, Decai Tang, Brandon J. Bethel

**Affiliations:** 1School of Law and Business, Sanjiang University, Nanjing 210012, China; tang_decai@sju.edu.cn or; 2China Institute of Manufacturing Development, Nanjing University of Information Science & Technology, Nanjing 210044, China; 3School of Marine Sciences, Nanjing University of Information Science & Technology, Nanjing 210044, China; 20195109101@nuist.edu.cn

**Keywords:** urbanization, Yangtze River Basin, environmental regulation efficiency, spatial econometrics, effects decomposition, carrying capacity of economic activities

## Abstract

Urbanization has been positioned as an important driving force for economic development. This article examines the impact of urbanization on environmental regulation efficiency (ERE) in the Yangtze River Basin (YRB). Based on a panel dataset of 97 cities in the YRB from 2005 to 2016, a spatial econometric model was used for analysis. Results show that the average ERE in the YRB is relatively low and manifests in the shape of a curved smile. The urbanization level of the permanent population is far lower than the average level of developed countries. However, the urbanization level is showing a steady growth trend. During this period, ERE in the Yangtze River middle, upper, and lower reaches was measured at 0.77, 0.58, and 0.52, respectively. The urbanization rate was measured at 0.59, 0.45, and 0.39, in the lower, middle, and upper reaches, respectively. When only considering population urbanization, the previously observed negative correlation between ERE and the Kuznets curve disappears. However, if the carrying capacity of economic activities is considered, the U-shaped relationship between urbanization rate and ERE returns. The environmental Kuznets curve is consequently verified. In addition, there is an inverted U-shaped nonlinear relationship between economic development and ERE. The results of this article show that there are unsustainable risks in the rapid pursuit of population urbanization. Only by improving the quality of urbanization and adapting the level of urbanization to the carrying capacity of resources and environment can we truly promote high-quality economic development. The article puts forward some suggestions to promote the green development of the economy.

## 1. Introduction and Background

In 2021, the Chinese Ministry of Finance and other departments jointly issued the Implementation Plan for Supporting the Establishment of a Horizontal Ecological Protection and Compensation Mechanism for the entire Yangtze River Basin [1], where the main goal was to improve the ecological environment of the Yangtze River Basin (YRB). Consequently, the National Development and Reform Commission issued the Key Tasks for New Urbanization and Urban−Rural Integration Development in 2021 [2]. This policy is geared towards advancing a new urbanization strategy centered on people, strengthening the construction of metropolitan areas, using central cities to drive neighboring cities, and accelerating the process of intra-urbanization. For example, Ge et al. [3] found that only by improving the efficiency of environmental control can the inclusive growth of local and neighboring economies be promoted. This result was based on the panel data of 281 cities in China from 2004 to 2016 and the two-region spatial Durbin model (SDM). Yu et al. [4] estimated the industrial ecoefficiency of 30 provinces in China from 2001 to 2015 based on the data envelopment analysis method. The authors verified the spatial convergence of industrial ecoefficiency through utilizing dynamic spatial econometrics. There is clear heterogeneity both in time and space. In the analysis of the influencing factors of environmental pollution, economic development and environmental pollution present an inverted N-shaped relationship. While industrial structure, technological research, and development all strongly affected environmental pollution, it was identified that foreign direct investment, by contrast, only minutely affected the environment [5]. The development of clean technologies should be strongly supported, because well-designed environmentally friendly policies could protect the environment and boost economic growth [6].

Tang et al. [7] argued that effective adjustment between land urbanization and urban ecological efficiency can promote sustainable economic development based on the mediating effect model. Land urbanization is not a simple inhibitory effect on ecological efficiency, but instead possesses a U-shaped relationship. The negative impact of land urbanization in northern prefecture-level Chinese cities on ecological efficiency is higher than that in southern cities. The establishment of urban agglomerations is an important sign of urbanization. During the process of urbanization, pollution emissions will be aggravated [8]. If the environment fails to receive adequate protection, economic development will be retarded while the process of urbanization would further aggravate environmental degradation. Fortunately, ecological innovation has been shown to stimulate the sustainable development of urbanization [9]. Human capital can also slow down the deterioration of the environment caused by the urbanization expansion [10]. The urbanization rate increases the domestic waste, and there is an inverted U-shaped relationship with the discharge of industrial pollutants [11].

Chen et al. [12] analyzed the impact of multidimensional urbanization on carbon emissions through a spatial panel model. The direct impact of population, land, and economic urbanization is positive, but the indirect impact of population urbanization is negative. Xing et al. [13] used data from 340 Chinese cities from 2000 to 2015 and employed a spatial panel measurement model to analyze the relationship between urbanization and the value of ecosystem services from the three dimensions of land, population, and economic urbanization. The land urbanization and economic urbanization of China can effectively improve the ecosystem services of surrounding cities. However, the urbanization of population has a negative impact on the surrounding ecological environment. Liang et al. [14] used geographic and time-weighted regression models to identify that reductions in environmental pollution are mainly due to the improvement of education and service industry levels, in addition to the increase of fiscal revenue and the popularization of the Internet. The urbanization rate, population agglomeration, economic development, industrial upgrading, urban construction, and transportation construction have all intensified environmental pollution. Urbanization has improved the environment in mountainous areas, but it has intensified pollution in plains and coastal areas. Therefore, a reasonable urban growth management policy is particularly important. An effective combination of urban development and environmental resource governance can avoid negative impacts on resources and the environment due to urban expansion [15]. Wu et al. [16] used a dynamic threshold panel model to analyze the interaction between urbanization and environmental pollution. Environmental pollution hinders the development of urbanization of the population but promotes the urbanization of the living environment. Rapid urbanization has indeed destroyed the structure and function of the ecosystem. Based on a study on the county level in Chongqing from 1997 to 2015, using time−space weighted regression (GTWR), it was found that population urbanization has exacerbated the deterioration of the natural ecosystem. The deterioration of the natural environment will also inhibit the sustainable development of urbanization [17].

Lv et al. [18] analyzed panel data from 30 Chinese provinces from 1997 to 2016 and concluded that in the process of rapid urbanization, energy efficiency needs to be improved and industrial structure be upgraded. China’s environmental pollution has the characteristics of spatial agglomeration with the high level of local urbanization, which will have side effects on the local environmental quality. Zhang et al. [19] used SDM to analyze the impact of urbanization on carbon dioxide emissions from two aspects of population urbanization and land urbanization. They found that while population urbanization has both positive and significant spatial spillover effects, these effects are virtually nonexistent for land urbanization. China’s population urbanization indicators have a large regional gap, but from 1995 to 2018, China’s ten largest urban agglomerations all experienced rapid urbanization development. With the exception of Beijing−Tianjin−Hebei and Sichuan−Chongqing urban agglomerations, the carbon storage of all others is in decline [20]. Population urbanization promotes carbon emissions [21]. Li et al. [22] used frontier stochastic models to estimate the energy efficiency of 30 Chinese provinces from 2003 to 2014 and analyzed the spatial dependence of energy efficiency using a spatial panel data model. The impact of urbanization on energy efficiency is not only reflected in the direct negative effects, but also in the spatial spillover effects brought about by the urbanization of neighboring areas. Excessive urbanization has indeed brought about an increase in pollution, mainly due to economic and human activities [23]. Although rapid urbanization has promoted economic development, it has seriously threatened the improvement of the ecological environment. From 2008 to 2017, the level of urbanization in China increased, but the efficiency of the ecological environment declined [24]. Regions and countries with higher levels of urbanization will pay more attention to coordinated development with the ecological environment. There is a certain coupling relationship and path dependence between urbanization and environment development. Studying the problems of urbanization and environmental regulation efficiency can provide decision-makers with reference for sustainable development and environmental protection [25].

The purpose of this article is to examine the impact of urbanization in the Yangtze River Basin on the efficiency of environmental regulations, as well as what changes will occur in the impact of such agglomeration on the efficiency of environmental regulations once resource carrying capacity is taken into account. Will urbanization be able to adequately safeguard the environment without compromising economic development if government policies change? This article’s main goals are to (1) study the impact of urbanization on environmental regulations in a systematic way and identify important contributing factors, and (2) construct appropriate static and dynamic spatial econometric models. It is desired that suitable recommendations are made to decision-makers so that the urbanization process may not only assure economic growth but also preserve environmental development.

This paper is structured as follows. After this introduction and background section, Section 2 presents the methodology, including the research area, econometric models, and data description. Section 3 analyzes the estimation results and effects decomposition. Section 4 examines the robustness check with three different methods, including core variable replacement, dynamic spatial econometric model, and time periods decomposition. In Section 5, we draw the main conclusions, and discuss the limitations of this paper and possible future research directions.

## 2. Methodology

### 2.1. Research Area

The YRB includes 17 provinces and two municipalities directly under the Central Government. Specifically, the mainstream flows through 11 provinces and cities, including Qinghai, Tibet, Sichuan, Yunnan, Chongqing, Hubei, Hunan, Jiangxi, Anhui, Jiangsu, and Shanghai. The tributaries flow through (except for the above 11 provinces and cities) Guizhou, Guangxi, Guangdong, Gansu, Shaanxi, Henan, Fujian, Zhejiang, and eight other provinces. The area of the YRB accounts for only about 20% of China but produces almost half of the country’s total economic output. The YRB flows through the three major urban agglomerations in China, namely the Chengdu−Chongqing, Yangtze River middle reaches, and the Yangtze River Delta urban clusters. Due to the availability of data, 97 cities in the YRB were finally selected as the study area (Figure 1).

### 2.2. Econometric Model

The spatial metrological analysis is the identification, measurement, and quantification of data location attributes. The characteristics of data location attributes are embodied in two aspects: spatial dependence and heterogeneity. The important reason for spatial dependence is the close proximity of geographical location and the mutual influence of human behavior activities. Spatial heterogeneity usually describes changes in spatial relationships, which can generally be illustrated by scatter plots. The quantification of spatial correlation is mainly achieved by location geographic information. This article mainly uses the Queen contiguity method to quantify the adjacency. As long as there is a common boundary or a common physical contact point between observation points, the space is adjacent and the weight value is 1, otherwise it is 0 [27].

The common forms of spatial measurement model mainly include spatial autoregressive model (SAR, originally called spatial lag model), spatial error model (SEM), and spatial Durbin model (SDM) [28]. The specific regression analysis are as follows (see Figure 2). First, pooled ordinary least squares (POLS) regression and Lagrange multiplier (LM) methods of SAR and SEM models are performed. The LM method can test the spatial interaction including spatial lag explained variables and spatial error autocorrelation [29]. Traditional LM and robust LM are based on the residuals of nonspatial models with spatial fixed effects, time-period fixed effects, and both spatial and time-period fixed effects. All models follow the chi-square distribution with a degree of freedom of 1. If the test result rejects the OLS null hypothesis and supports the SAR or SEM model, SDM should be used for estimation, because the SDM model includes both the spatial lag explained variable (WY) and the spatial lag explanatory variable (WX) [30]. To further test whether SDM can be converted to a simpler SEM or SAR, the likelihood ratio (LR) test, and Wald test should be used for verification. Under the premise that the null hypothesis is rejected, the spatial error model and the spatial lag model must be rejected; the SDM model can produce better fitting effects [28].

This article used Elhorst MATLAB routines to estimate spatial panel data, including mixed, individual, time-period, and both individual and time-period fixed effects [31]. The fixed effects model is based on the averaging procedure proposed by Baltagi [32] and also known as the direct method. If the model includes both individual and time-period effects, the parameter estimates may be biased. Therefore, Lee and Yu [33] proposed a bias correction procedure of parameter estimation for the direct method. The final effect comparison and selection is determined by the LR test. If the probability value of LR is greater than 0.05, Hausman needs to be further performed. If the *p*-value of the LR and Wald test is less than 0.05, the SDM fixed-effects model with bias correction estimates should be selected. The individual fixed effects should be selected if the probability value is less than 0.05 when comparing individual fixed effects and mixed effects. The time-period fixed effects models should be chosen if the probability value is less than 0.05 when comparing time-period fixed effects and mixed effects models. When comparing the individual fixed effects and both individual and time-period fixed effects, the latter should be chosen if the probability value is less than 0.05.

This paper constructed spatial panel data of 97 cities in the YRB from 2005 to 2016, then used the data to analyze the effects of population urbanization on the environmental regulation efficiency in the YRB. The spatial econometric model was constructed as follows:(1)$$Y_{it}=\alpha +\rho \overset{n}{\underset{j=1}{\sum }}W_{ij}Y_{jt}+\beta X_{it}+\theta \overset{n}{\underset{j=1}{\sum }}W_{ij}X_{ijt}+\mu_{i}+\xi_{t}+\varepsilon_{it}$$
(2)$$\varepsilon_{it}=\;\lambda M\varepsilon_{t}+{\;u}_{it}$$
where *Y* is the dependent variable, *X* is the explanatory variable, $W_{i}$ represents the *i*th row of the spatial weight matrix *W*, $\mu_{i}\;$and $\xi_{t}$ represent the optional spatial effect and time effect, respectively, and ε represents the disturbance term, *i* = 1, 2, …, *n*, *t* = 1, 2,..., *t*. *W* and *M*, respectively, represent the spatial weight matrix of the dependent variable and the disturbance term. *ρ* is the coefficient of the spatial lag term of the dependent variable, *θ* is the spatial autocorrelation coefficient of the explanatory variable, *β* and *θ* represent parameter vectors, *β* reflects the influence of the explanatory variable on the dependent variable, and λ is the spatial correlation coefficient of the error term. The model can test the following two hypotheses: H0: θ=0; and H0: θ+ρβ=0; if θ=0,SDM will degenerate into the spatial lag model SAR. If θ=−ρβ, SDM will degenerate into the spatial error model. If ρ=0, and λ=0,SDM will degenerate into OLS.

To avoid bias, this paper constructed SDM and SAR models and a nonspatial measurement panel model to verify the impact of urbanization in the YRB on the efficiency of environmental regulation. These models were finally set up as follows:(3)$$ERE_{it}=\alpha +\rho \overset{n}{\underset{j=1}{\sum }}W_{ij}ERE_{jt}+\beta^{\prime }CORE_{it}+\theta^{\prime }\overset{n}{\underset{j=1}{\sum }}W_{ij}CORE_{ijt}+\beta^{\prime \prime }CONT_{it}\;+\theta^{\prime \prime }\overset{n}{\underset{j=1}{\sum }}W_{ij}CONT_{ijt}+\mu_{i}+\xi_{t}+\varepsilon_{it}$$
(4)$$ERE_{it}=\alpha +\rho \overset{n}{\underset{j=1}{\sum }}W_{ij}ERE_{jt}+\beta^{\prime }CORE_{it}+\beta^{\prime \prime }CONT_{it}+\mu_{i}+\xi_{t}+\varepsilon_{it}$$
(5)$$ERE_{it}=\alpha +\beta^{\prime }CORE_{it}+\beta^{\prime \prime }CONT_{it}+\mu_{i}+\xi_{t}+\varepsilon_{it}$$

The SDM model is displayed in Equation (3) and the SAR model is shown in Equation (4). Equation (5) presents the nonspatial panel data model. $W_{ij}$ stands for the weight matrix W between cities i and j. The queen weight matrix is applied to all the spatial models. ERE is the dependent variable. CORE represents the core explanatory variables, including urbanization rate and its square, to examine the nonlinear relationship. CONT is the control variables. Other variables are set in the same way as benchmark Equation (1).

The parameter estimation of the nonspatial model can represent the marginal effect of urbanization on the efficiency of environmental regulation, but the coefficient of the spatial lag explanatory variable cannot be used to measure the spatial spillover effect under the SDM model [34]. Direct and indirect effects can be used to measure the degree of influence of explanatory variables on the dependent variables. The direct effect is the regression coefficient in the usual sense, that is, the impact of the local urbanization level on the efficiency of local environmental regulation. The reason why the direct effects of explanatory variables are different from their coefficient estimates is the existence of the feedback effects. The feedback effects are the effects passing through surrounding regions and then back to affect this region itself. The feedback effects are the direct effects minus the coefficients of the SDM model. The indirect effect is the spatial spillover effect, that is, the effect of the urbanization level in the region on the efficiency of environmental regulation in neighboring regions [28]. The total effect is the cumulative effect, which is the sum of the direct effect and the indirect effect [30].

### 2.3. Data Source

This paper used DEA-SolverPro13.1 software (SAITECH, Tokyo, Japan) and selected the super-efficiency non-radial slacks-based data envelopment analysis model (SE-SBM) to calculate the ERE [26]. If the result is greater than 1, the environmental regulation is effective. If the result is less than 1, the environmental regulation is ineffective. If the score is between 0.5 and 1, ERE is weakly ineffective, and if the score is less than 0.5, ERE is strongly ineffective. ERE is the dependent variable.

Core explanatory variables are urbanization rate and its square. Urbanization is a process in which nonagricultural industries gather in urban areas and rural populations gather in urban areas with the development of industrialization. It is measured by the proportion of permanent residents in the total population of the region. The urbanization rate is an important indicator reflecting the level of urbanization. To examine whether there is a Kuznets curve, this article also takes the square of the urbanization rate as a core explanatory variable. The data comes from statistical yearbooks of provinces and cities, government work reports, and statistical bulletins of each city.

Control variables are mainly selected from three aspects of the economy, society, and environment. The five economic indicators are as follows. The level of economic development (GDP) is measured with the total real GDP and calculated based on the nominal GDP and the index of the previous year. These data are collected from the China National Knowledge Internet (CNKI) [35] and statistical yearbooks of each province and city. This article also selects the square of GDP to verify the nonlinear relationship between GDP and ERE. The proportion of the secondary industry in GDP, measured by the output value of the secondary industry divided by GDP, comes from the statistical bulletins and statistical yearbooks of each city. Investment in fixed assets, which mainly reflects the scale, structure and development speed of fixed asset investment, comes from CNKI and the China City Statistical Yearbook (CCSY). The level of economic openness is measured by the actual use of foreign capital. The data comes from the statistical bulletins and city statistical yearbooks of various provinces and cities. The social indicators mainly include employment rate, education level, and scientific research level. The employment level is measured by dividing the total number of employees in the whole society by the permanent population of the city. The total number of employment data in the whole society comes from the CCSY. The permanent population data comes from the EPS data platform, China Population and Employment Statistical Yearbook, and statistical yearbooks of various cities. Education level is measured by the number of students in ordinary colleges and universities. The data comes from the CCSY. Research level is measured by the number of employees in scientific research, technical services, and geological exploration. The data comes from the CCSY. The environmental indicators mainly include precipitation, water resources, and forestry output. The data of average annual precipitation was compiled from the Water Resources Bulletin (official document issued by Water Resources Department of each province), Climate Bulletin (official document issued by Meteorological Bureau of each province and city), and CCSY. Total water resources refer to the amount of water produced on the ground and underground. The data comes from the CNKI data platform, Water Resources Bulletin, and statistical yearbook. The forestry output value data comes from the CNKI platform, CCSY, City Statistical Bulletin, and statistical yearbooks of various cities.

Among the variables, 11 observations of the number of students in ordinary colleges and universities have a value of 0, and five observations of the actual use of foreign capital have a value of 0. To take the logarithm and conform to the positive definite matrix assumption, the variables of 0 were taken as 1 person and RMB 0.01, respectively, to obtain the smallest possible positive value. Other missing values were improved by averaging or interpolation, and the total missing values do not exceed 5%. Table 1 presents the descriptive statistics of each variable.

## 3. Results

### 3.1. Spatial Effect Test

The concept of spatial autocorrelation was first proposed by Moran [36]. The Moran’s index is generally between 0 and 1. More than 0 indicates that there is a positive spatial autocorrelation. Less than 0 indicates that there is a negative spatial autocorrelation. If the value is close to 0, the spatial distribution is random and there is no spatial autocorrelation. The global Moran’s index represents the spatial agglomeration of the entire spatial sequence, and the local Moran’s index represents the spatial agglomeration of a certain area. In this paper, Geoda software was adopted to conduct univariate Moran’s index and univariate local Moran’s index on the environmental regulation efficiency of the 97 cities. Both Moran’s indices are 0.231. Later, up to 999 permutation tests were performed, and the probability value was always less than 0.05, indicating that a positive spatial correlation exists. The Moran’s index is the slope of the regression line in the scatter plot.

The Lagrange multiplier (LM) test was used to determine whether there are spatial lag variables or spatial error variables that affect the dependent variables in the model. As shown in Table 2, among the four nonspatial models, the *p*-values of the traditional spatial lag LM test and spatial error LM test are 0, and all passed the test at the 1% level. This article examined the robust spatial lag LM test and spatial error LM test, and it was found that the spatial error LM estimates of the four models all passed the test. However, the spatial lag LM estimates of the mixed effect model and the individual fixed effect model failed the test. The spatial lag LM of the time-period fixed effect model and fixed effect model with both the individual and time-period effects passed the test at 5% level. Therefore, the null hypothesis that there are no spatial lag and spatial error effects is rejected. To further judge whether the null hypothesis is significant, this article performed an LR test. If the *p*-value of the LR test estimates is 0, this indicates that the null hypothesis should be rejected. The results of these tests prove that the fixed effects model with spatial and time-period effects must be extended, that is, the two-way fixed effects model both with individual and time-period should be selected [37].

### 3.2. ERE and Urbanization in the YRB

From 2005 to 2016, the average super-efficiency value of environmental regulations in 97 cities in the YRB was 0.66, and the overall ERE was not high. The urbanization level of the permanent population in the YRB is 46.76%. The current urbanization rate of China’s permanent population is 53.7%, which is far lower than the average level of 74% in developed countries. The average urbanization level of developing countries whose per capita income is similar to that of China has reached about 53%. In 2016, the efficiency of environmental regulation in the YRB improved, reaching an average of 0.70, and the level of urbanization also increased to 54.38, an increase of nearly 16% compared with 2005. As shown in Figure 3, the overall environmental regulatory efficiency shows a “smile curve” shape. Since 2012, the efficiency has improved, and the level of urbanization has shown a steady increase as well.

From 2005 to 2016, the average efficiency of environmental regulations in the middle reaches of the Yangtze River, the upper reaches of the Yangtze River, and the lower reaches of the Yangtze River were measured at 0.77, 0.58, and 0.52, respectively. The urbanization rate in the lower, middle, and the upper reaches of the Yangtze River were measured at 0.59, 0.45, and 0.39, respectively. In 2016, the urbanization rates of the upper, middle, and lower reaches of the Yangtze River were 0.48, 0.53, and 0.64, respectively. In the National New Urbanization Plan (2014–2020), the target of permanent population urbanization rate is to reach about 60%. The lower reaches of the Yangtze River, especially the Yangtze River Delta urban agglomerations, mainly use the spatial spillover effect of the Shanghai Free Trade Zone to promote urbanization and new rural construction. After 2011, the efficiency of environmental regulations and urbanization rates in the lower reaches of the Yangtze River showed a simultaneous growth trend. Figure 4 shows that the efficiency of regional environmental regulation and the level of urbanization development in the YRB are significantly different.

### 3.3. Spatial Econometric Estimation

According to the nonspatial model with both individual and time-period fixed effects, the urbanization of the permanent population has a negative and significant relationship with ERE. The Kuznets curve in between has not been verified. The more developed the economy, the higher the efficiency of environmental regulation. The economic development, to a certain level, shows a negative relationship with the efficiency of environmental regulation, but this result is not significant. The development of the secondary industry and the level of investment in fixed assets have a restraining effect on the efficiency of environmental regulations, and the impact of the secondary industry is even more obvious. The level of foreign investment can promote the efficiency of environmental regulation. From the perspective of social factors, increasing the employment rate has a positive effect on the efficiency of environmental regulation, but the level of education and scientific research inhibits the efficiency of environmental regulation. From the perspective of resource factors, cities with abundant water and forestry resources have higher ERE.

From the perspective of the spatial models, to test the null hypothesis of whether the SDM can be reduced to a simple spatial error model, this paper performed the LR and Wald tests. The results of the LR test (25.2501, *p* = 0.0214) and Wald test (22.7676, *p* = 0.0446) show that the null hypothesis should be rejected. Similarly, to test the null hypothesis of whether the SDM can be reduced to a simple spatial lag model, the results of the LR test (34.2269, *p* = 0.0011) and Wald test (32.631, *p* = 0.0019) rejected the null hypothesis as well. Consequently, the individual and time-period fixed effects SDM model with the biased correction estimates were utilized. To make comparisons easier, the time-period effects SAR and SDM, the individual and time-period fixed effects SAR, and the SDM results without the biased estimates are listed in Table 3. There, it can be observed that the results of the estimated biased correction model and estimated model without biased correction are different, but the difference is not significant. The results of the spatial model and the nonspatial model are quite different. In the SDM with biased correction, urbanization and ERE have a significant negative correlation, but the Kuznets curve still cannot be verified. Unlike the nonspatial model, the conclusion that scientific research inhibits the improvement of environmental regulation efficiency cannot be effectively verified. In terms of environmental resources, the conclusion that precipitation promotes the efficiency of environmental regulations cannot be effectively verified.

### 3.4. Effects Decomposition

This article decomposes the spatial effects of urbanization on the efficiency of environmental regulation into direct, indirect, and total effects (Table 4). The direct impact of the urbanization of the permanent population on the efficiency of environmental regulation is negative, and there is no Kuznets curve. The increase in the level of local urbanization will also have a negative impact on the environmental regulation of neighboring cities. Although the result is not significant, the negative impact on the surrounding cities will eventually be fed back to inhibit the environmental regulation efficiency on one’s own city. In general, urbanization has inhibited the improvement of the efficiency of environmental regulations. Considering other influencing factors, the improvement of the economic level can promote the improvement of the efficiency of local environmental regulations, but the economic development, to a certain extent, has a certain inhibitory effect on the impact of environmental regulations. The development of the secondary industry and the direct impact of fixed asset investment on the efficiency of environmental regulation are negative, but the spatial spillover effect is not significant. The direct and overall effects of foreign direct investment are both positive, which can promote the improvement of the efficiency of local environmental regulations. It has a positive cumulative effect on environmental regulations, and the feedback effect is also positive. From the perspective of social factors, increasing employment can not only promote the efficiency of local environmental regulations, but also ultimately produce a positive feedback effect by influencing surrounding cities. The more educated people in a city there are, the more negative effects there will be on the local and adjacent cities. The higher the education level, the more developed the economy, and the efficiency of environmental regulations will be more negatively affected. For example, Managi and Jena [38] used a panel data model to demonstrate that the improvement of education level effectively reduced the production of SO_2_ but increased NO_2_ and suspended particles in the air. Balaguer and Cantavella [39] verified that education increases the population and initially increases CO_2_ emissions, but when it reaches a certain level, the improvement of education level can reduce carbon emissions. It has not yet been demonstrated that scientific research investment will have a negative impact on the efficiency of environmental regulations. Zafar et al. [40] argued that technological innovation reduces environmental quality. Only through the use of green technology, attracting foreign investment, and expanding the scale of the city can the environmental quality be improved [39]. In terms of resources, cities with sufficient rainfall can have a positive pulling effect on local and surrounding environmental regulations. The more abundant the total amount of water resources and forestry resources, the more efficiently the local environmental regulations can be promoted. In cities with abundant water resources, drainage facilities and wastewater treatment are more effective [41]. Forestry is very important in mitigating climate change and carbon sequestration in forests. Forestry development can achieve ecological benefits, thereby improving the efficiency of local environmental regulations [42].

## 4. Robustness Check

### 4.1. Transform Core Variable Method

In order to test the robustness of the model results, we replaced the core explanatory variables with economic agglomeration (EA), that is, the spatial agglomeration of economic activities [43]. To better describe the spatial density and distribution of economic activities, and accurately reflect the carrying capacity of economic activities per unit geographic area, the concept of economic agglomeration was introduced. The degree of economic agglomeration is quantified by the ratio of nonagricultural output (the sum of the output of the secondary and tertiary industries) to the urban administrative area [44]. At the same time, considering the nonlinear possibility of economic agglomeration and the efficiency of environmental regulation, the square of economic agglomeration (EA2) is also adopted as one of the core explanatory variables, replacing the original square of the urbanization of the permanent population. Other variables remain unchanged. First, the spatial autocorrelation is tested. In this paper, the traditional LM and robust LM tests were performed on nonspatial models including the mixed effects model, individual effects model, time-period fixed effects model, and fixed effects model with both individual and time-period effects. The LR test was used for different model effects selection. All the choices rejected the original hypothesis, that there were no spatial lags and spatial errors. Finally, a fixed effects model with both individual and time-period effects was selected. The conclusions are consistent with the previous estimation results. Furthermore, LR and Wald tests were used to select SAR, SEM, and SDM, and the final result estimates all pointed to the bias-corrected SDM model at a significant level of 5% (Table 5).

This article replaced the core variable urbanization rate and its square with economic agglomeration and its square, and the explanatory and control variables remained unchanged. According to the results in Table 6, we also reported the results of SAR and SDM, and found that the degree of economic agglomeration still inhibits the efficiency of environmental regulations. There is no nonlinear relationship. The nonspatial model with time and individual fixed effects also reported the same result.

Table 7 reflects the direct, indirect, total, and feedback effects of each explanatory variable on the efficiency of traditional environmental regulation. The direction of the coefficients estimates of direct effects, indirect effects, and overall effects remains basically unchanged. Interestingly, in the robustness check, the relationship between economic agglomeration and environmental regulation efficiency is not a simple negative relationship, but a nonlinear relationship. The higher the degree of economic agglomeration, the lower the efficiency of environmental regulation will be. The efficiency of environmental regulation is gradually improving until the degree of economic agglomeration reaches a certain turning point and there is a U-shaped relationship between the two variables. The environmental Kuznets curve hypothesis of economic growth and ERE has been verified. Economic growth will promote the improvement of the efficiency of environmental regulation, but when economic growth reaches a certain turning point, the efficiency of environmental regulation will decline with economic growth. The reason for this conclusion is that the urbanization indicator of the permanent population only considers the demographic factor, while economic agglomeration not only considers the economy and population, but also considers the carrying capacity of economic activities. Urbanization is a multidimensional concept. It is a process in which a society dominated by agriculture is gradually transformed into an urban society dominated by nonagricultural industries, such as secondary and tertiary industries. The process of urbanization includes not only the agglomeration of people, but also the agglomeration of economy [45,46].

### 4.2. Dynamic Spatial Econometric Model

To further verify the impact of urbanization on the efficiency of environmental regulation, a dynamic SDM was built based on the original equation. This dynamic model was built to consider the results of the initial state of the explained variable on the entire model. The core explanatory variables are still the urbanization rate and its square, and other control variables are consistent with the benchmark model. The specific dynamic SDM model is set as follows:(6)$$ERE_{it}=\tau ERE_{i,t-1}+\rho^{\prime }\overset{n}{\underset{j=1}{\sum }}W_{ij}ERE{\;}_{j,t-1}+\rho^{\prime \prime }\overset{n}{\underset{j=1}{\sum }}W_{ij}ERE_{jt}+\beta^{\prime }CORE_{it}+\theta^{\prime }\overset{n}{\underset{j=1}{\sum }}W_{ij}CORE_{ijt}\;+\beta^{\prime \prime }CONT_{it}+\theta^{\prime \prime }\overset{n}{\underset{j=1}{\sum }}W_{ij}CONT_{ijt}+\mu_{i}+\xi_{t}+\varepsilon_{it}$$

In addition to the SDM results, the article also provides SAR results for comparison (Please see Table 8). The results show that urbanization has a certain inhibitory effect on the improvement of environmental regulation efficiency. This is consistent with the interpretation of the benchmark model.

### 4.3. Decomposition of Time Periods

The *Central Economic Work Conference,* held in 2012, emphasized that China should actively and steadily promote urbanization and improve the quality of urbanization and use urbanization as a starting point to improve the quality and efficiency of economic growth. In the process of China’s urbanization development, there are unsustainable risks. Therefore, the *Central Economic Work Conference* proposed that new urbanization should be developed as “intensive, smart, green, and low-carbon”, and urbanization should be compatible with the carrying capacity of resources and the environment. Urbanization requires not only quantitative expansion, but also qualitative improvement. Therefore, to further verify the temporal and spatial evolution of the impact of the urbanization process in the YRB on environmental regulations, the article demonstrates the two time periods from 2005 to 2012 and 2013 to 2016. Table 9 below shows estimation results of the static SDM and dynamic SDM models. From 2005 to 2012, the urbanization rate increased rapidly, but it inhibited the efficiency of environmental regulations. The reason was that only quantity was emphasized at that time, and quality was not emphasized. After 2012 was also an important time during the *Twelfth Five-Year Plan* period (2011–2015). China no longer only emphasized quantity in the process of urbanization but began to pay attention to the matching of urbanization and the carrying capacity of resources and the environment. Quality urbanization has begun to help improve the efficiency of environmental regulations.

## 5. Conclusions and Discussion

In the analysis of the effects of population urbanization rate on environmental regulation efficiency in 97 cities in the YRB from 2005 to 2016, the overall efficiency of environmental regulation in the YRB is not high, and there are large differences between different regions. The overall development of ERE presents a smiling curve trend. The urbanization rate of the population is showing a continuous growth trend. Compared with developed countries, the urbanization rate still has a lot of room for improvement. Overall, the urbanization rate and the efficiency of environmental regulations have not shown a trend of simultaneous growth. After 2011, however, the efficiency of environmental regulations in the lower reaches of the Yangtze River has shown a trend of simultaneous growth. This paper used Moran’s index and Elhorst LM test to verify the existence of the spatial autocorrelation relationship of ERE in the YRB. It was verified that the increase in population urbanization inhibits the increase in the efficiency of environmental regulation. Promoting a shift of the population from being agriculture-focused to one that engages in more activities will likely have a negative effect on environmental regulations. A U-shaped relationship between economic agglomeration and the efficiency of environmental regulation was also observed. After decomposing the spatial effects and fully considering the impact of urbanization, economy, society, and resource environment on the efficiency of environmental regulations, it was found that these factors will not only affect the local area, but also affect the surrounding cities, and the impact on the surrounding cities will eventually even feed back into itself. In terms of direct effects, urbanization and environmental regulation efficiency are negatively correlated, and the environmental Kuznets curve hypothesis has not been verified. The development of the secondary industry and the investment in fixed assets have a negative impact on the local environment. Increasing the employment rate can effectively improve the efficiency of environmental regulation. The increase in FDI will bring about capital growth, innovative products, and technologies, alleviate the problem of rising costs caused by environmental regulation, and ultimately improve the efficiency of environmental regulation. In terms of resources, the more abundant the water resources and forestry resources, the higher the efficiency of local environmental regulations. The spatial effect of the urbanization rate is negative, but not significant. In the overall effect, the relationship between the two variables is still a significant negative correlation.

Urbanization can promote the structural transformation of the economy, and it is also an important driving force for the structural transformation of the economy. However, to pursue higher regional GDP and taxation, local governments will have excessive preference for industrial enterprises and physical investment when formulating and implementing economic policies. Therefore, public investment in human capital, such as education, is seriously insufficient. However, with the development of China’s economy, economic restructuring has gradually changed to take the people’s pursuit of a better life as the basic law. In 2013, China again began to promote urbanization again. Although urbanization is developing very rapidly, China remains below the average level of developed countries as seen from the perspective of population urbanization. As the YRB plays a vital role in China’s economic development, urbanization is an important driving force for expanding domestic demand. In the process of promoting urbanization, too much attention is paid to speed and quantity, which brings many unsustainable risks to economic development. Urbanization cannot only be measured by population size. It should also focus on matching urbanization with the carrying capacity of resources and the environment. Only by gradually shifting from the pursuit of quantity to the pursuit of high-quality urbanization can real sustainable development be achieved.

This article does have limitations. The degree of urbanization in developed countries is much higher than that in China, and there are large differences in basic national conditions. Perhaps the research results do not apply to those developed countries at present. However, there are still many developing countries, such as countries along the Belt and Road Initiative and other low-income countries. Urbanization is one of the important driving forces of economic development. The development of urbanization will inevitably lead to the development of industrialization, and then the development of industrialization will, to a large extent, give rise to environmental problems. Well-designed policies will play a very important guiding role. Therefore, in the process of urbanization, how to coordinate the economy and the environment, as well as how to exert the space radiation effect of cities, are issues that need to be continuously discussed. In the context of global warming and carbon neutrality, maintaining sustainable economic development without sacrificing the ecological environment is an issue worthy of study. Despite the limitations of the article, the results do suggest that simply pursuing excessive growth will bring certain damage to the ecological environment. In the process of economic growth, resource carrying capacity needs to be considered, and reasonable policies can effectively improve ERE. In future research, we will focus on the agglomeration phenomenon and further explore the transmission mechanism on sustainable economic development. Furthermore, we will explore how this agglomeration phenomenon is conducive to the realization of the great vision of global carbon neutrality.

In the formulation of policies, we should not blindly aim to increase the urban population but should instead correctly guide population flows in the process of urbanization. The government should therefore fully consider the carrying capacity of economic and social activities. Moreover, the government should advocate the formulation of a new type of urbanization, stabilize employment, expand the opening to the outside world, realize industrial transformation and upgrading, and strengthen the protection of natural resources to promote sustainable green economic development.

## Figures and Tables

**Figure 1 ijerph-18-09105-f001:**
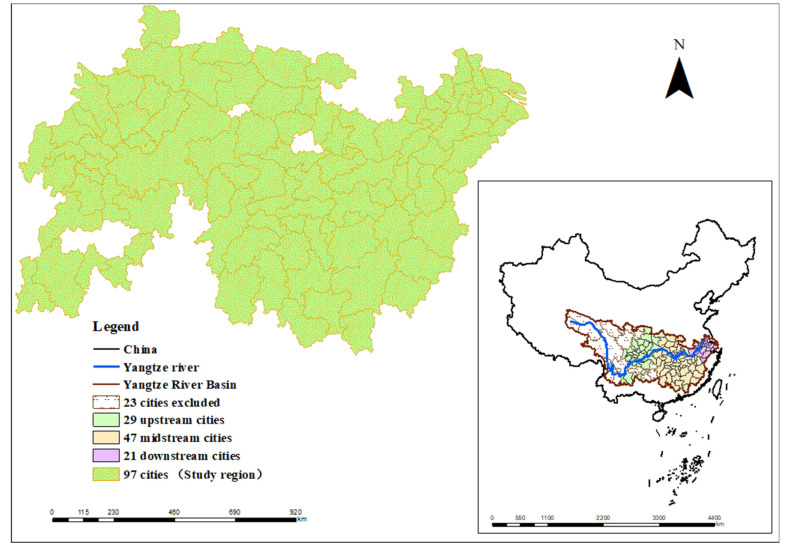
Study region of 97 cities in the YRB. Reprinted with permission from ref. [26]. 2021 Zhang, Q. et.al.

**Figure 2 ijerph-18-09105-f002:**
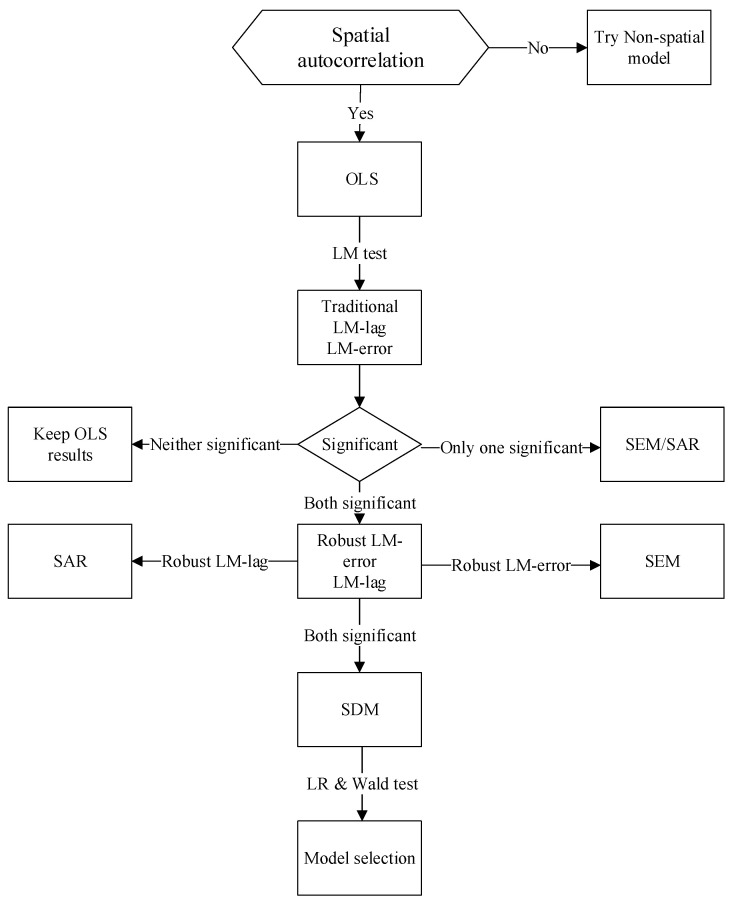
Flow chart of methodology.

**Figure 3 ijerph-18-09105-f003:**
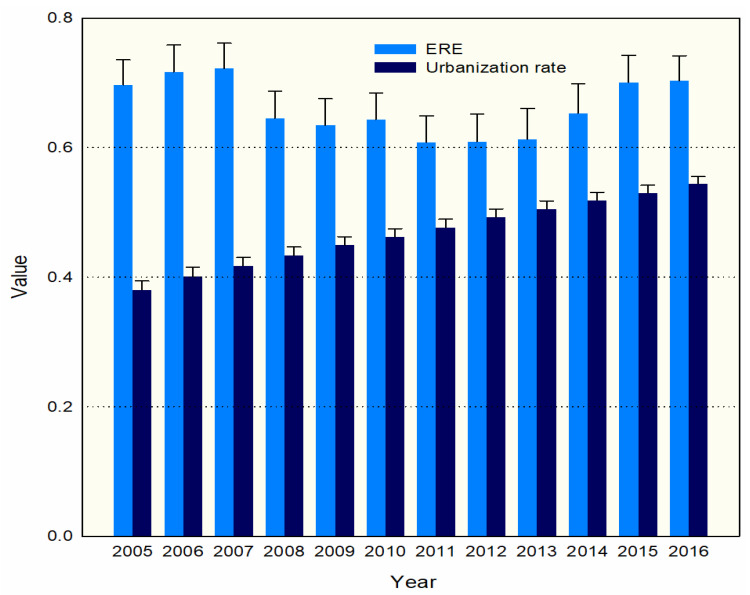
Overall ERE and urbanization rate grouped bar chart with error bars.

**Figure 4 ijerph-18-09105-f004:**
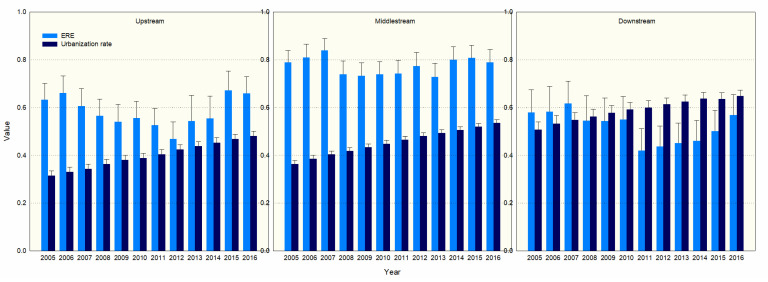
Regional ERE and urbanization rate grouped bar charts with error bars.

**Table 1 ijerph-18-09105-t001:** Descriptive statistics of each variable.

Variable	Sub Index	Unit	Mean	Std Dev	Min	Max
Dependent variable						
Environmental Regulation Efficiency	ERE		0.66	0.41	0.01	2.68
Core explanatory variables						
Urbanization rate	UR		0.47	0.14	0.15	0.89
Square of urbanization rate	UR2		0.24	0.14	0.02	0.79
Control variables						
Real GDP	GDP	CNY 10^3^ million	17.23	27.37	0.99	269.24
Square of real GDP	GDP2	CNY 10^3^ million	1045.09	4691.77	0.98	72,492.75
Share secondary sector in GDP	SEC		0.49	0.09	0.19	0.75
Fixed asset investment	INV	CNY 10^3^ million	11.57	15.15	0.36	173.61
Foreign direct investment	FDI	USD 10^3^	847.06	2002.98	0.00	18,513.79
Employment rate	EMP		0.60	0.08	0.36	0.86
Students enrolment by regular colleges and universities	EDU	10^3^ persons	9.35	16.96	0.00	96.64
Scientific personnel	SCI	10^3^ persons	1.16	3.44	0.05	81.00
Precipitation	PRE	mm	1284.79	424.73	248.70	3012.00
Water resource	WR	10^3^ million m^3^	0.98	0.92	0.04	6.63
Output of forestry	OFOR	CNY 100 million	11.42	12.92	0.30	104.76

**Table 2 ijerph-18-09105-t002:** LM and LR test in panel data mode.

	OLS_1	OLS_2	OLS_3	OLS_4
LM lag test	147.6386 ***	159.8063 ***	123.1966 ***	133.9748 ***
LM error test	164.4008 ***	178.8973 ***	125.2793 ***	137.9622 ***
Robust LM lag test	2.9319 *	2.5811	5.6917 **	4.9251 **
Robust LM error test	19.6942 ***	21.6722 ***	7.7744 ***	8.9125 ***
Time fixed effect	Mixed effects	No	Yes	Yes
Individual fixed effect		Yes	No	Yes
R^2^	0.2879	0.2762	0.2751	0.2603
*N*	1164	1164	1164	1164
		**Statistics**	**Degree of Freedom**	***p*-value**
LR test	the joint test1	53.1963	97.0000	0.9999
	the joint test2	67.9338	12.0000	0.0000
	the joint test3	54.5100	12.0000	0.0000

Note: *: significance at 10% level, **: significance at 5% level, ***: significance at 1% level. T statistic values are in parentheses. The joint test1 is the comparison between individual fixed effects and mixed effects. Joint test2 is the comparison between time-period fixed effects and mixed effects models. Joint test3 is the comparison between the individual fixed effects and both individual and time-period fixed effects.

**Table 3 ijerph-18-09105-t003:** Estimation results of SAR and SDM.

Variable	Spatial Model					Nonspatial Model
	SAR	SDM	SAR	SDM	SDM *	OLS
UR	−1.121343 **	−0.868615	−1.388635 **	−1.055698 **	−1.049110 *	−1.623942 ***
	(−2.140199)	(−1.609745)	(−2.536761)	(−1.97196)	(−1.870835)	(−2.889892)
UR2	0.498699	0.128359	0.774529	0.303359	0.297132	0.929304
	(0.901078)	(0.223475)	(1.343369)	(0.531936)	(0.4974)	(1.570098)
GDP	0.003564 *	0.003859 *	0.004545 **	0.005026 **	0.005008 **	0.005256 **
	(1.792999)	(1.905058)	(2.129204)	(2.420443)	(2.302624)	(2.399552)
GDP2	−0.000007	−0.000007	−0.000008	−0.000007	−0.000007	−0.000011
	(−1.04157)	(−0.974175)	(−1.128412)	(−1.094817)	(−1.029991)	(−1.578603)
SEC	−0.492762 ***	−0.538400 ***	−0.508013 ***	−0.568194 ***	−0.568366 ***	−0.522027 ***
	(−3.661702)	(−3.708376)	(−3.61607)	(−3.910851)	(−3.734638)	(−3.62277)
INV	−0.004656 ***	−0.00493 ***	−0.004251 **	−0.004593 ***	−0.004591 **	−0.004654 **
	(−2.670486)	(−2.806153)	(−2.301393)	(−2.601316)	(−2.482303)	(−2.455122)
FDI	0.000054 ***	0.000049 ***	0.000041 ***	0.000035 **	0.000035 **	0.000051 ***
	(3.690511)	(3.350218)	(2.665936)	(2.368392)	(2.248143)	(3.229076)
EMP	0.003214 ***	0.002888 **	0.00248 **	0.002123 *	0.002112 *	0.003289 ***
	(3.007344)	(2.569144)	(2.203022)	(1.89044)	(1.795462)	(2.847909)
EDU	−0.004067 ***	−0.003087 ***	−0.004319 ***	−0.003287 ***	−0.003262 ***	−0.005959 ***
	(−3.903455)	(−2.634261)	(−3.898363)	(−2.795489)	(−2.648839)	(−5.261515)
SCI	−0.004977	−0.003697	−0.006429 *	−0.005054	−0.004990	−0.006944 *
	(−1.349625)	(−1.012048)	(−1.664229)	(−1.38585)	(−1.306226)	(−1.750974)
PRE	0.000075 **	0.000063 **	0.000066 **	0.000050	0.000050	0.000087 ***
	(2.477535)	(2.062301)	(2.044955)	(1.625882)	(1.530375)	(2.634072)
WR	0.092805 ***	0.106022 ***	0.100500 ***	0.115384 ***	0.115664 ***	0.100889 ***
	(4.957687)	(5.574846)	(5.135429)	(6.069064)	(5.808092)	(5.028344)
OFOR	0.416390 ***	0.524953 ***	0.386561 ***	0.491668 ***	0.493988 ***	0.331205 **
	(2.990328)	(3.52565)	(2.637407)	(3.302432)	(3.167672)	(2.201434)
W*dep.var.	0.360968 ***	0.380952 ***	0.390090 ***	0.391967 ***	0.413869 ***	
	(11.321567)	(10.967925)	(12.354214)	(11.393733)	(12.255799)	
Time fixed effect	Yes	Yes	Yes	Yes	Yes	Yes
Individual fixed effect	No	No	Yes	Yes	Yes	Yes
R^2^	0.4000	0.4175	0.4417	0.4584	0.4608	0.2603
N	1164	1164	1164	1164	1164	1164

Note: W queen contiguity matrix; SDM*: SDM with biased corrections. *: significance at 10% level, **: significance at 5% level, ***: significance at 1% level. T statistic value in parentheses.

**Table 4 ijerph-18-09105-t004:** Results of direct, indirect, total, and feedback effects.

Variable	Direct Effects	Indirect Effects	Total Effects	Feedback Effects
UR	−1.187567 **	−1.933853	−3.121420 *	−0.138457
	(−2.114549)	(−1.193672)	(−1.740746)	
UR2	0.425533	1.799653	2.225186	0.128401
	(0.71301)	(1.09154)	(1.222679)	
GDP	0.005269 **	0.004712	0.009981	0.000261
	(2.41145)	(0.693871)	(1.339751)	
GDP2	−0.000009	−0.000031	−0.000041 *	−0.000002
	(−1.263731)	(−1.488344)	(−1.70818)	
SEC	−0.554179 ***	0.113173	−0.441007	0.014187
	(−3.64697)	(0.290208)	(−1.03078)	
INV	−0.004608 **	−0.000448	−0.005056	−0.000017
	(−2.412479)	(−0.072518)	(−0.733609)	
FDI	0.000040 **	0.000064	0.000104 *	0.000005
	(2.527858)	(1.344878)	(1.961673)	
EMP	0.002420 **	0.004279	0.006699 *	0.000308
	(2.058294)	(1.294914)	(1.844125)	
EDU	−0.003700 ***	−0.006260 **	−0.009961 ***	−0.000438
	(−3.08582)	(−2.097398)	(−3.172495)	
SCI	−0.006389	−0.019205	−0.025594	−0.001399
	(−1.628775)	(−1.218166)	(−1.466419)	
PRE	0.000064 *	0.000225 **	0.000289 ***	0.000014
	(1.866091)	(2.348774)	(2.691658)	
WR	0.109131 ***	−0.097590	0.011540	−0.006533
	(5.169375)	(−1.627428)	(0.174594)	
OFOR	0.448176 ***	−0.772391 *	−0.324214	−0.045812
	(2.933089)	(−1.747516)	(−0.677519)	

Note: *: significance at 10% level, **: significance at 5% level, ***: significance at 1% level. T statistic value in parentheses.

**Table 5 ijerph-18-09105-t005:** Results of LM test, LR test, and Wald test for robustness check.

	OLS_1	OLS_2	OLS_3	OLS_4
LM lag test	154.2764 ***	164.7856 ***	126.6116 ***	135.0475 ***
LM error test	178.7808 ***	193.6510 ***	128.5727 ***	140.8812 ***
robust LM lag test	1.3139	0.6876	5.8770 **	4.1408 **
robust LM error test	25.8183 ***	29.5530 ***	7.8381 ***	9.9745 ***
Time fixed effect	Mixed effects	No	Yes	Yes
Individual fixed effect		Yes	No	Yes
R^2^	0.2777	0.2619	0.2721	0.2533
*N*	1164	1164	1164	1164

Note: *: significance at 10% level, **: significance at 5% level, ***: significance at 1% level.

**Table 6 ijerph-18-09105-t006:** Estimation results of SAR and SDM of robustness check.

Variable	Spatial Model					Nonspatial Model
	SAR	SDM	SAR	SDM	SDM with Biased Corrections	OLS
EA	−0.004813 ***	−0.005433 ***	−0.004474 ***	−0.005191 ***	−0.005169 ***	−0.005184 ***
	(−3.549800)	(−3.780953)	(−3.121316)	(−3.548390)	(−3.373492)	(−3.522880)
EA2	0.000006	0.000006	0.000007	0.000007	0.000007	0.000008
	(1.008517)	(0.983670)	(1.125182)	(1.224815)	(1.152413)	(1.317521)
W*dep.var.	0.356966 ***	0.387976 ***	0.389174 ***	0.405958 ***	0.427790 ***	
	(11.158719)	(11.249092)	(12.294312)	(11.948586)	(12.834784)	
Time fixed effect	Yes	Yes	Yes	Yes	Yes	Yes
Individual fixed effect	No	No	Yes	Yes	Yes	Yes
Control variables	Yes	Yes	Yes	Yes	Yes	Yes
R^2^	0.3986	0.4202	0.4370	0.4569	0.4592	0.2533
N	1164	1164	1164	1164	1164	1164

Note: W queen contiguity matrix. *: significance at 10% level, **: significance at 5% level, ***: significance at 1% level. T statistic value in parentheses.

**Table 7 ijerph-18-09105-t007:** Decomposition of effects of SDM with biased corrections.

	Direct Effects	Indirect Effects	Total Effects	Feedback Effects
EA	−0.005579 ***	−0.005915	−0.011494 **	−0.000410
	(−3.732977)	(−1.394990)	(−2.478650)	
EA2	0.000009	0.000031	0.000040 *	0.000002
	(1.529136)	(1.595942)	(1.890666)	
GDP	0.012051 ***	0.011179	0.023231 **	0.000750
	(4.072384)	(1.208876)	(2.265081)	
GDP2	−0.000021	−0.000088 *	−0.000108 **	−0.000006
	(−1.475729)	(−1.791406)	(−2.044476)	
SEC	−0.839403 ***	0.129228	−0.710175 *	0.005426
	(−6.400008)	(0.346530)	(−1.758495)	
INV	−0.008126 ***	−0.000385	−0.008512	−0.000048
	(−3.565018)	(−0.054822)	(−1.049845)	
FDI	0.000037 **	0.000079	0.000116 **	0.000006
	(2.281266)	(1.535462)	(2.034522)	
EMP	0.000315	0.003566	0.003882	0.000292
	(0.269920)	(1.047491)	(1.046538)	
EDU	−0.005937 ***	−0.005813 *	−0.011751 ***	−0.000393
	(−5.069239)	(−1.957956)	(−3.621090)	
SCI	−0.007704 *	−0.025525	−0.033229 *	−0.001716
	(−1.957216)	(−1.475988)	(−1.738456)	
PRE	0.000068 **	0.000276 ***	0.000345 ***	0.000019
	(1.989841)	(2.647210)	(2.938534)	
WR	0.103216 ***	−0.129570 **	−0.026354	−0.009714
	(5.053589)	(−2.070353)	(−0.380299)	
OFOR	0.372071 **	−0.859746 **	−0.487674	−0.058650
	(2.415898)	(−2.001990)	(−1.064965)	

Note: *: significance at 10% level, **: significance at 5% level, ***: significance at 1% level. T statistic value in parentheses.

**Table 8 ijerph-18-09105-t008:** Estimation results of dynamic SDM and SAR.

Variable	SDM	SAR
ERE_-1_	−0.065782 **	−0.065365 ***
	(−2.383926)	(−4.676587)
W*ERE_-1_	0.002630	0.000104
	(0.059095)	(0.647455)
UR	−0.994762 *	−1.284771 **
	(−1.835912)	(−2.546129)
UR2	0.183559	0.601021
	(0.318459)	(1.112380)
W*dep.var.	0.442318 ***	0.404929 ***
	(11.988172)	(11.786150)
Time fixed effect	Yes	Yes
Individual fixed effect	Yes	Yes
Control variables	Yes	Yes
R2	0.5089	0.4923
N	1067	1067

Note: W queen contiguity matrix. *: significance at 10% level, **: significance at 5% level, ***: significance at 1% level. T statistic value in parentheses.

**Table 9 ijerph-18-09105-t009:** Estimation results of SDM during different time periods.

	2005–2012		2013–2016	
	Static SDM	Dynamic SDM	Static SDM	Dynamic SDM
ERE_-1_		−0.016542		−0.186691 ***
		(−0.453621)		(−3.968360)
W*ERE_-1_		0.104028		−0.075905
		(1.604838)		(−0.855634)
UR	−1.326845 *	−1.663589 **	−6.406104 ***	−5.159839 ***
	(−1.926628)	(−2.307987)	(−3.753073)	(−3.454759)
UR2	0.768663	0.883500	5.346160 ***	4.383812 ***
	(1.003637)	(1.094815)	(3.289669)	(2.985536)
W*dep.var.	0.254047 ***	0.306038 ***	0.248646 ***	0.221987 ***
	(5.467574)	(6.106238)	(3.793809)	(2.860299)
Time fixed effect	Yes	Yes	Yes	Yes
Individual fixed effect	Yes	Yes	Yes	Yes
Control variables	Yes	Yes	Yes	Yes
R2	0.4739	0.4977	0.5498	0.5289
N	776	679	388	291

Note: W queen contiguity matrix. *: significance at the 10% level, **: significance at the 5% level, ***: significance at the 1% level. T statistic value in parentheses.

## Data Availability

The datasets used in this research are available upon request.

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
