# Peer review of "Impact of Urbanization on the Environmental Regulation Efficiency in the Yangtze River Basin Based on the Empirical Analysis of Spatial Econometrics"

_ijerph, 2021, doi:10.3390/ijerph18179105_

Round 1

Reviewer 1 Report

The comments from the previous review were mostly covered. I add only a few more that appeared in the new text.

Page 3, line 20:
Separate quote from the word: is declining(Ouyang et al. 2021).

Page 3, line 120:
has declined(Yao et al. 2021b).

Page 5, line 189:
Remove blank before colon: was constructed as follows:.

Lines 410 and 412:
I expected SO2, NO2 and CO2 and not SO2, NO2 and CO2.

General comment:
Please avoid long sentences. Rewrite all sentences that are excessively long.

Author Response

Many thanks.

Reviewer 2 Report

After the first revision, the manuscript "Impact of Urbanization on the Environmental Regulation Efficiency in the Yangtze River Basin-Based on the Empirical Analysis of Spatial Econometrics" has been significantly improved and now publication can be warranted in IJERPH. The authors well replied to my suggestions and modified the text according to my requests.

However, some remarks regard:

  1. several incomplete sentences, with the lack of the verb. They should be completed with an appropriate verb or they should be tied with appropriate punctuation marks to the previous sentence. Examples are at lines:
  • 406 “The more educated people in a city, the more negative effects on the local and neighboring cities”;
  • 433 “In order to better describe the spatial density and distribution of economic activities and accurately reflect the carrying capacity of economic activities per unit geographic area”;
  • 467 “The higher the degree of economic agglomeration, the lower the efficiency of environmental regulation”.

A further check should be done overall the text.

  1. References are not complete. For example, there are no references for:
  • Implementation Plan for Supporting the Establishment of a Horizontal Ecological Protection and Compensation Mechanism for the entire Yangtze River Basin (line 36);
  • Key Tasks for New Urbanization and Urban-Rural Integration Development (line 39);
  • DEA-SolverPro13.1 software (line 232);
  • China National Knowledge Internet (CNKI) and statistical yearbooks (line 249);
  • Water Resources Bulletin, Climate Bulletin (line 268);

A further check should be done overall the text.

Author Response

Many thanks.

Reviewer 3 Report

The revision has improved the caliber of the paper. But still there are major issues around the paper/study.

The type of paper was not stated, it does not read like a 'Research Article" at best 'Case Report' seems to be the correct type as the paper solely focuses on a unique case study. 

The literature background needs to be improved. The following would help but the issue is also not to have a dedicated section to report the literature findings to inform the rationale of the study and the study approach. Either have a dedicated Literature Review section or expand the content and change the Introduction section to Introduction and Background section. 

Incorporate the following article where you talk about the need for green technology/innovation/development.

Aldieri, L., Carlucci, F., Vinci, C. P., & Yigitcanlar, T. (2019). Environmental innovation, knowledge spillovers and policy implications: A systematic review of the economic effects literature. Journal of Cleaner Production, 239, 118051.

The findings are still not transferable to any other context than the case study or at best the same country. Discuss this issue and its limitations in the Discussion section. 

A careful language check is needed. 

Reference formatting is wrong, and should follow journal's guidelines and template. 

Author Response

Many thanks.

This manuscript is a resubmission of an earlier submission. The following is a list of the peer review reports and author responses from that submission.

Round 1

Reviewer 1 Report

Dear authors,
I would like to express my congratulation for the paper presented for the consideration of publication in International Journal of Environmental Research and Public Health. In the paper the impact of urbanization on environmental regulation is analised in deteail, making a picture of the current and future situation. The paper is well written and the structure is well organized. I would only like to point out some minor changes that I detected during the reading. 
(1) Lines 42-43:
Check location of the comma
(2) Lines 60-63:
Please, rephrase the statement because it is too long
(3) Line 67 and 90:
Please introduce a blank space <<Chen et al.(2020)>> <<Lv et al.(2020)>>
(4) Lines 95-99:
Please, rephrase the statement because it is too long
(5) Lines 104-105:
Please, add some statements between headings of sections
2. Methodology
2.1. Research area
(6) Line 167:
Please, remove the blank space before colon <<following model:>>
(7) 
Please, try to organize better the data in the tables. In many cases is not comfortable to read and extract the ideas from the figures
(8)
Please, use subscript index with chemical atoms such as NO2, CO2 or SO2

Reviewer 2 Report

The article "Impact of Urbanization on the Environmental Regulation Efficiency in the Yangtze River Basin-Based on the Empirical Analysis of Spatial Econometrics" is really notable. It discusses the effects of urbanization on the environmental regulation efficiency in the Yangtze River Basin from a spatial econometric point of view. The application of Moran’s Index and Elhorst LM test enables to verify the existence of their spatial autocorrelation. Then, the spatial Durbin model (SDM) with both spatial and time-period fixed effect is applied to deepen the relationship between the two variables. The results show that the spatial effects, decomposed into direct, indirect and overall effects, and the impact of urbanization, economy, society, and resource environment on the efficiency of environmental regulation will affect both the local area and the surrounding cities, as well as the impact on the surrounding cities eventually feeds back to itself.

Some additional remarks:

  • The abstract should be reorganized showing only the econometrics indices, functions and tests that will be applied in the study, with the methodological approach and the possible results the authors expect to obtain. It is not recommended to write explicitly the numerical values of the results which, on the other hand, would make more sense in the “conclusions”.
  • The sentence at line 42 “Following an analysis performed by Ge et al. (2020) where, based on the panel data of 281 cities in China from 2004 to 2016 and the two-region spatial Durbin model (SDM), found that only by improving the efficiency of environmental 44 control can the inclusive growth of local and neighboring economies be promote” should be reorganized.
  • At line 90, it is suggested to specify better the case study of Lv et al. Writing “the panel data of provinces” is is very vague. Which are? How many? where?
  • It is suggested to move to line 200 the beginning of the sentence “Control variables are 199 mainly selected from three aspects of economy, society, and environment. The economic 200 indicators are as follows.”
  • In section “2. Methodology”, a new sub-paragraph must be added, that should describe the methodology, also with the use of a flow chart or graphical diagram, because the results present procedural steps, each one characterized by a specific goal.
  • It is suggested to reorganize the following sections:
  1. Section “3. Results” should be renamed as “3. Results and discussion”;
  2. Section “3.3. Results of spatial econometric estimation” shoul be renamed as “3.3. Spatial econometric estimation”;
  3. Section “4. Robustness check” should become a subsection of paragraph 3: “3.5 Robustness check”;
  4. Section “5. Conclusions and Discussion” should contains only “5. Conclusions”.
  • Sub paragraphs 3.4 and the new 3.5 should be more detailed on the results obtained, moving the considerations on their goals supported by the scientific literature references in the new methodology subparagraph.
  • The notes at “Table 2. LM and LR test in panel data mode” should contain the meaning of the “*” and of the parentheses.
  • “Table 4. Results of direct, indirect, overall effects and feedback effects.” should be recalled in the text.
  • “Table 6. Decomposition of effects for robustness check” probably should contain “economic agglomeration” instead of “urbanization” as stated at line 395 “…we replaced the core explanatory variables with economic agglomeration…”.

Reviewer 3 Report

Referee report for the manuscript entitled “Impact of Urbanization on the Environmental Regulation Efficiency in the Yangtze River Basin-Based on the Empirical Analysis of Spatial Econometrics”

The manuscript focuses on a relevant topic and fits well within the broad scope of the journal. Nevertheless, it contains some significant limitations. These limitations include the lack of methodological calibre, novelty, and significance of the findings as well as applicability and relevance of the findings in other contexts. Furthermore, the study reported in the manuscript is not enough of an advance and of enough impact for the journal. Considering these key limitations, I do not believe the manuscript to be considered for publication in this leading journal of the field.

Some of the issues behind this decision are highlighted below:

  • Examining the impact of urbanization on environmental regulation efficiency in Yangtze River Basin is not an adequate aim for a study.
  • The findings should be transferable to other country contexts.
  • There is not a research question or hypothesis to address or validate/falsify.
  • There is not a sufficient literature background.
  • The critical issue of the manuscript is that it is missing a sound Discussion section. Without such section the manuscript does not clearly address the ‘so what’ question for research, policy and practice.
  • The last issue is that a careful language editing is needed to improve the readability and flow of the ideas in the manuscript.